# Cross-serotype protection against group A Streptococcal infections induced by immunization with SPy_2191

Pooja Sanduja [1,6,8], Manish Gupta [2,8], Vikas Kumar Somani [2,7], Vikas Yadav [1], Meenakshi Dua[3], Emanuel Hanski [4], Abhinay Sharma [4], Rakesh Bhatnagar [5,9] & Atul Kumar Johri [1,9 ✉]

Group A *Streptococcus* (GAS) infection causes a range of diseases, but vaccine development is hampered by the high number of serotypes. Here, using reverse vaccinology the authors identify SPy_2191 as a cross-protective vaccine candidate. From 18 initially identified surface proteins, only SPy_2191 is conserved, surface-exposed and inhibits both GAS adhesion and invasion. SPy_2191 immunization in mice generates bactericidal antibodies resulting in opsonophagocytic killing of prevalent and invasive GAS serotypes of different geographical regions, including M1 and M49 (India), M3.1 (Israel), M1 (UK) and M1 (USA). Resident splenocytes show higher interferon-γ and tumor necrosis factor-α secretion upon antigen re-stimulation, suggesting activation of cell-mediated immunity. SPy_2191 immunization significantly reduces streptococcal load in the organs and confers ~76-92% protection upon challenge with invasive GAS serotypes. Further, it significantly suppresses GAS pharyngeal colonization in mice mucosal infection model. Our findings suggest that SPy_2191 can act as a universal vaccine candidate against GAS infections.

[1] School of Life Sciences, Jawaharlal Nehru University, New Delhi 110067, India. [2] BSL-3 Unit, Molecular Biology and Genetic Engineering Laboratory, School of Biotechnology, Jawaharlal Nehru University, New Delhi 110067, India. [3] School of Environmental Sciences, Jawaharlal Nehru University, New Delhi 110067, India. [4] Department of Microbiology and Molecular Genetics, The Institute for Medical Research–Israel-Canada(IMRIC), Faculty of Medicine, The Hebrew University of Jerusalem, Jerusalem, Israel. [5] Banaras Hindu University, Varanasi, Uttar Pradesh, India. [6] Present address: Division of Infectious Diseases, Department of Pediatrics, Boston Children's Hospital, Harvard Medical School, Boston, MA, USA. [7] Present address: Division of Oncology, Washington University School of Medicine, St. Louis, MO, USA. [8] These authors contributed equally: Pooja Sanduja, Manish Gupta. [9] These authors jointly supervised this work: Rakesh Bhatnagar, Atul Kumar Johri. ✉ email: akjohri14@yahoo.com

*S*treptococcus pyogenes or GAS is a human pathogenic bacterium. It causes a range of suppurative diseases (pharyngitis, impetigo), invasive diseases [necrotizing fasciitis, streptococcal toxic shock syndrome (STSS)] and post-streptococcal sequel [Acute rheumatic fever (ARF), rheumatic heart disease (RHD), glomerulonephritis]. Annually, GAS causes 616 million cases of pharyngitis, 18.1 million severe cases and 517,000 deaths worldwide[1]. GAS is ninth leading infectious bacteria in the estimate of mortality and falls with measles, *Haemophilus influenza* type b and hepatitis B. Further, GAS causes high morbidity and mortality mostly in low and middle-income countries. GAS pathogenicity is underestimated due to lack of data from developing countries (South-Asian and Sub-Saharan African countries).

The M protein of GAS is a surface-exposed protein with a highly variable N-terminal region that forms the basis of different serotyping in GAS[2]. More than 220 serotypes of GAS are prevalent in different geographical regions[3]. Prevalence of a serotype also changes in few years with time in different regions[4,5]. The M protein is a major virulence factor of GAS that helps in adhesion and invasion of bacteria to epithelial cells and also in evading the host innate immune response due to its anti-phagocytic function[6–8]. Few vaccine preparations like 26-valent, 30-valent and J8 were made based on the M-protein, are currently in phase I or II clinical trials. Additionally, various other subunit vaccines like C5a peptidase, GAS carbohydrate and serum opacity factor, have also shown promising results, however no clinical trials were conducted related to these preparations[9–14]. The progress in development of an effective vaccine against GAS is further impeded due to serotype diversity in different geographical areas, antigenic variation within serotype and cross-reacting antibodies causing auto-immune disorders like ARF and RHD[2,3,15,16]. Currently, antibiotics like penicillin and cephalosporins among others are in use to combat various GAS diseases. However, antibiotic resistance developed by some GAS clinical isolates against macrolides and tetracyclines in various geographical regions, has led to a worldwide concern[17]. Till date, regardless of a high demand globally, no vaccine has been licensed against GAS infections.

Genome sequences of various pathogenic bacteria and viruses are available for the past two decades and have been exploited immensely in vaccine development. One approach, which was found to be highly successful to identify universally applicable vaccine candidates, is reverse vaccinology. It was first tested on serogroup B meningococcus[18]. Reverse vaccinology coupled with comparative genomics, proteomics, and bioinformatics allow reducing the number of pre-clinical candidates to be analyzed for immunogenicity[19–21]. It has been established that a successful vaccine candidate must be conserved, immunogenic, either surface exposed or secretory and should be well expressed[22]. Importantly, universal vaccine candidates must protect against serotypes prevalent in different geographical areas. Based on reverse vaccinology approach, we predicted a total of 147 genes as universal GAS vaccine candidates. We further validated the in silico analysis by exploring the distribution profile of these predicted genes in non-sequenced Indian GAS strains. Among these, 52 genes were present in all the prevalent GAS serotypes of Indian origin[21]. In the current study, the available 45 recombinant sera previously generated against these 52 and the other reported genes[20,21] are screened for their role in adherence and invasion. Among those that are found to be involved in adherence are subsequently checked for their exposure from the surface of GAS serotypes of Indian origin. Only one candidate, SPy_2191 tests as a potential vaccine candidate in the mouse model against five prevalent and invasive GAS serotypes from India, Israel, UK and USA. Importantly, this finding highlights SPy_2191 as a promising universal vaccine candidate, in providing significant protection against the globally prevalent and invasive GAS serotypes in different geographical areas.

## Results

**Inhibition of adherence and invasion.** For effective vaccination, the potential vaccine candidate must be surface exposed, involved in invasion and adherence[23–26]. Out of 52 previously predicted vaccine candidates, 45 sets of immune and preimmune mouse antisera, generated against recombinant surface/secretory proteins of GAS (Supplementary Table 1)[20,21] were used to investigate whether the corresponding surface/secretory proteins had any role in adherence or invasion. Initially, GAS serotype M49 that caused outbreaks in India and USA was used for this study as this serotype was found to be most invasive[27,28]. We found that 18 out of 45 antisera inhibited GAS adhesion by ≥75%; however, only seven antisera abolished invasion by ≥80% (Table 1). We found SPy_2191 antisera inhibited adherence and invasion by 89 and 93%, respectively (Table 1). Preimmunized mice antisera for each recombinant GAS protein were used as a control.

**Selection of surface-exposed antigen.** One of the criteria for effective vaccination against any pathogen involves surface exposure of an antigen and its accessibility by specific host-generated antibodies[29]. To narrow down our focus in subsequent studies on the development of a broad-range GAS vaccine, we employed all the 18 antisera (inhibiting adhesion) to confirm the exposure of their respective target antigen on Indian GAS M49 cell surface using flow cytometry analysis. Data were analyzed using the mean fluorescence intensity (MFI) obtained in both immunized and preimmunized antisera. Candidates showing >2.4-fold increase in MFI as compared to preimmune controls were considered significantly surface exposed (Fig. 1a). Out of 18 protein candidates, 10 were found to be significantly exposed on GAS cell surface (Fig. 1a, Table 1).

**In silico analysis and selection of SPy_2191.** Out of the 10 potential surface-exposed GAS vaccine candidates, nine did not show significant inhibition of invasion and therefore were not considered for further experiments (Table 1). Only SPy_2191-showed significant inhibition of both adhesion and invasion. Additionally, in overlay diagram, GAS M49 incubated with SPy_2191-immunized antisera had significantly higher MFI in comparison to GAS alone and GAS M49 incubated with pre-immune antisera (Fig. 1b), suggesting SPy_2191 is surface exposed. Therefore SPy_2191 was selected for subsequent studies. Our BLAST analysis revealed that the SPy_2191 coding sequence was highly conserved across various GAS serotypes of developed and developing countries, with >98% similarity in all the GAS genome sequences available in the NCBI database.

**SPy_2191 elicit a humoral immune response.** The SPy_2191 antigen was expressed in *E. coli* BL21 (DE3) cells as His$_6$-tagged recombinant protein and purified to >95% purity by Ni-NTA affinity chromatography (Supplementary Figs. 1–4). Mice immunization was performed as illustrated in Fig. 2a. The SPy_2191-specific serum IgG titres were measured at day 14 after priming, followed by booster immunizations (day 28 and 42) by indirect ELISA (Fig. 2b). Compared to primary immunization, the total serum IgG endpoint titers after the 1st and 2nd booster immunizations were found to be ~3.5 and ~5-fold higher, respectively (p < 0.0001) (Fig. 2b). Our data related to high IgG titres (induced by SPy_2191 vaccination) suggested a resilient humoral immune response.

**Table 1 Neutralization assay showing inhibition of GAS M49 adherence and invasion to HEp-2 cells.**

| NCBI reference no | Protein annotation | % adhesion inhibition | % invasion inhibition |
|---|---|---|---|
| [a]NP_268656 | [c]ABC transporter substrate-binding protein | **77.6** | 19.1 |
| [a]NP_268747 | [c]ABC transporter metal binding protein (lipoprotein) | **81.4** | 29.0 |
| [a]NP_269961 | [c]Surface lipoprotein | **75.3** | 40.8 |
| [a]NP_269968 | [c]Laminin adhesion | **99.8** | 36.1 |
| [a]NP_269203 | [c]Extracellular hyaluronate lyase | **79.8** | 0.95 |
| NP_269743 | [c]Esterase | **78** | 24.0 |
| [a]NP_269944 | [c]Streptokinase A/ Streptokinase A precursor | **75.8** | 65.7 |
| NP_269981 | [b]ATP-binding cassette transporter-like protein | **77.8** | **89.9** |
| [a]NP_269984 | [b]Foldase PrsA | **91.7** | **95** |
| [a]NP_270099 (This study) | Hypothetical protein SPy_2191/Transglycosylase SLT domain family protein | **89** | **93** |
| NP_269045 | [b]Hypothetical protein SPy_0836 | **96** | **96.9** |
| NP_269051 | [c]Hypothetical protein SPy_0843 | **99.6** | 73.3 |
| [a]NP_270005 | [b]Dipeptidase | **88.3** | **99.7** |
| [a]NP_270119 | [c]Serine protease | **89.2** | 23.1 |
| [a]NP_269570 | [b]Hypothetical protein SPy_1492 | **86** | **84.6** |
| [a]NP_269402 | [b]cAMP factor | **97** | 36.9 |
| [a]NP_268542 | [b]ABC transporter lipoprotein | **78.9** | 15.7 |
| [a]NP_269268 | [b]Acid phosphatase/ phosphotransferase | **99.7** | **86.4** |

For this purpose 18 antisera generated against surface proteins were selected.
[a]Represent protein candidates showing high relative expression in M49/IND as compared to M1/IND GAS.
[b]Out of these 18, eight candidates were rejected as they were not found to be surface exposed (Fig. 1).
[c]From the remaining 10 candidates, nine candidates, though found to be involved in adherence as well as surface exposed but not showed significant inhibition in invasion therefore were not selected.
Only one candidate i.e., NP_270099 showed inhibition in adherence and invasion as well as found to be surface exposed (Fig. 1), therefore was selected for further study. Bold numbers: More than 75% inhibition in adherence and invasion.

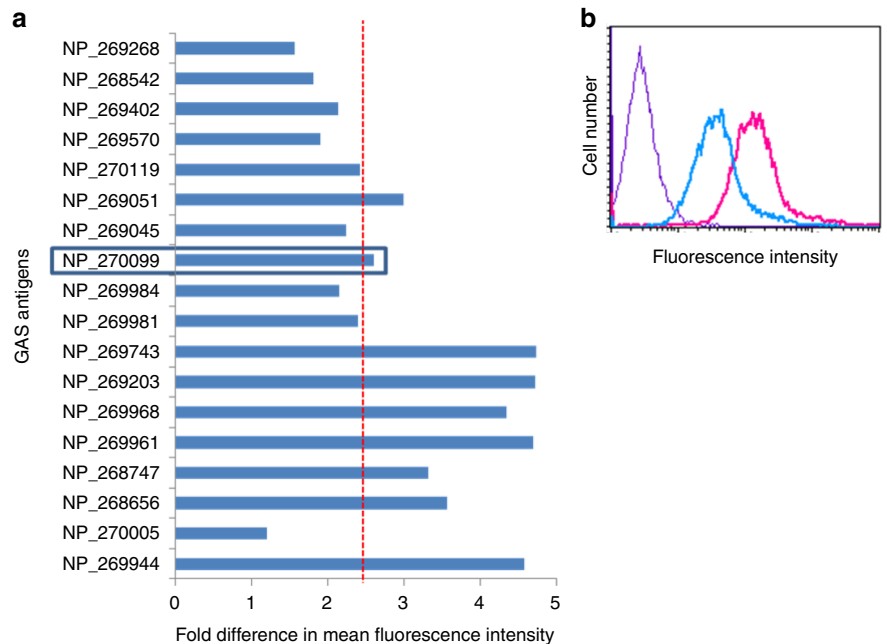

**Fig. 1 Surface localization of potential vaccine candidates on GAS M49 surface. a** Surface exposure analysis of 18 selected candidates. GAS M49 cells were incubated with protein-specific polyclonal antibodies and surface localization was analyzed by flow cytometry using FITC-conjugated secondary goat anti-mouse IgG. Mean fluorescence intensity (MFI) obtained for each immunized serum (I) was normalized with the MFI of preimmunized sera (PI) to measure the fold difference. Serum candidates showing >2.4-fold difference in MFI (represented by red dotted line) were considered significant. **b** Histogram depicting SPy_2191 surface exposure. In the overlay diagram, purple line represents control M49 GAS cells without antisera, blue line represents cells with preimmunized antisera and red line represents cells incubated with SPy_2191 antisera. Experiment was repeated two times and data shown here are a representative experiment.

**SPy_2191 -specific serum antibodies mediate GAS opsonophagocytic killing.** Conservation of SPy_2191 sequences across different GAS serotypes led us to hypothesize that SPy_2191 could trigger protective immunity in a serotype-independent manner. Hence, to ascertain the biological activity of the SPy_2191 -specific IgG antibodies, sera was tested for complement mediated opsonophagocytic killing of different GAS serotypes by whole blood bactericidal assay. For this purpose,

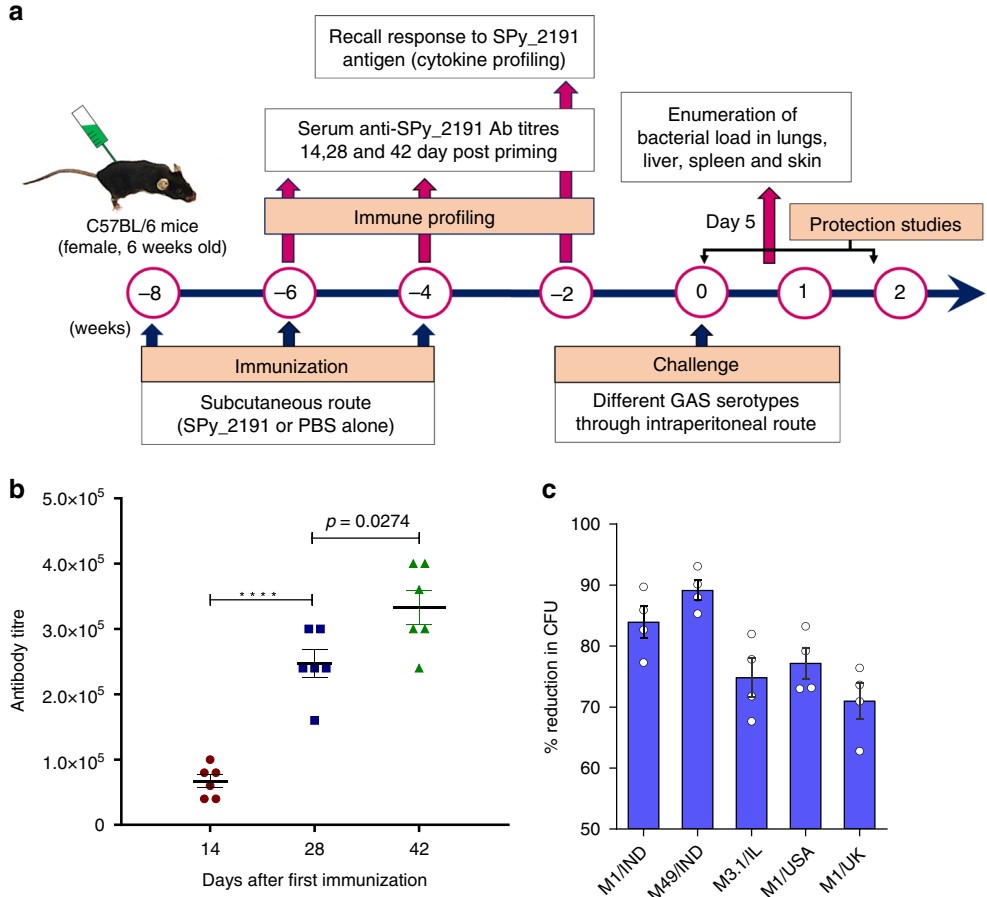

**Fig. 2 SPy_2191 induces humoral immune response. a** Immunization schedule in C57BL/6 mice. **b** SPy_2191 specific total IgG endpoint titres in sera samples ($n = 6$ mice) collected 2 weeks after priming (day 14) and booster immunizations (day 28 and 42), was measured by Indirect ELISA. One-way ANOVA followed by Tukey's multiple-comparison test was employed for calculating the significant difference. Data are presented as mean values ± SEM **c** Opsonophagocytic activity of SPy_2191 specific antibodies containing sera ($n = 4$), was evaluated for bactericidal assay, against various GAS serotypes, viz. M1/IND, M49/IND, M3.1/IL, M1/USA, and M1/UK. Percent reduction of CFU by SPy_2191-immunized sera was estimated in comparison to PBS-immunized mice sera, used as control. Data are presented as mean values ± SEM. ****$P < 0.0001$.

PBS-immunized mice sera were used as control. Importantly, all the tested GAS serotypes (used in this study) were found to be prevalent, caused invasive diseases in various geographical areas like USA, Australia, New Zealand, and UK in the past decade[27,30–33]. Remarkably, in the presence of SPy_2191 -specific antibodies the mean reduction in CFUs of different GAS serotypes was observed between 70 and 90 %, with highest reduction was found in case of M49/IND (~90 %) followed by M1/IND (~84 %) and M1/USA (~77 %) (Fig. 2c). Likewise, in case of M3.1/IL and M1/UK GAS serotypes, ~75 and 71 % reduction in CFUs were observed (Fig. 2c).

**SPy_2191 evoked Th1-baised immune response**. To monitor the efficacy of SPy_2191 vaccination on the development of helper T cell, the memory-recall response of ex vivo cultured splenocytes was assessed by measuring the level of different cytokines released in the culture supernatant after antigen stimulation. Two weeks after the final booster (day 42), mice were euthanized; splenocytes were isolated and stimulated with SPy_2191 antigen for 72 h. Overall, a superior cytokine response was observed in SPy_2191-vaccinated mice as compared to PBS-immunized mice, which produced low amount of cytokines upon antigen-stimulation (Fig. 3). Among all the analysed cytokines, Th1-specific IFN-γ and TNF-α level were found to be ~27- and ~15-fold higher in case of SPy_2191-vaccinated mice as compared to PBS-treated

mice (Fig. 3a, b). In case of IL-2 no significant difference was observed between both the groups (Fig. 3c). In contrast to the strong Th1 response, Th2-specific IL-6 levels were found to be ~7.5-fold significantly higher in the culture supernatant of SPy_2191-vaccinated mice as compared to PBS-treated mice (Fig. 3d). However, the levels of IL-4 and IL-10 cytokine levels were found to be statistically insignificant under similar condition (Fig. 3e, f). Additionally, stimulation with SPy_2191 antigen triggered a robust secretion of Th17-specific proinflammatory cytokine like IL-17A ($P < 0.001$; Fig. 3g).

**In situ protection and expanded serotype coverage by SPy_2191 vaccination**. We further assessed protective efficacy of SPy_2191 upon challenge with GAS serotypes from different geographical areas. Two weeks after the final booster, SPy_2191 and PBS vaccinated mice were challenged intraperitoneally with the $LD_{50}$ dose of different GAS serotypes (Supplementary Table 2). Four days post-challenge, mice were sacrificed aseptically and the total bacillary load in the liver, lungs, spleen, and skin was enumerated. Remarkably, in case of all the serotypes, a significant decrease i.e., ~3–5 $Log_{10}$ GAS CFU load was observed in all the organs of SPy_2191-immunized mice as compared to the PBS-treated mice ($p < 0.01$). This decrease in total bacillary load in case of each organ was found to be >99% (Fig. 4a–e). Direct protection to streptococcal infection and extended GAS

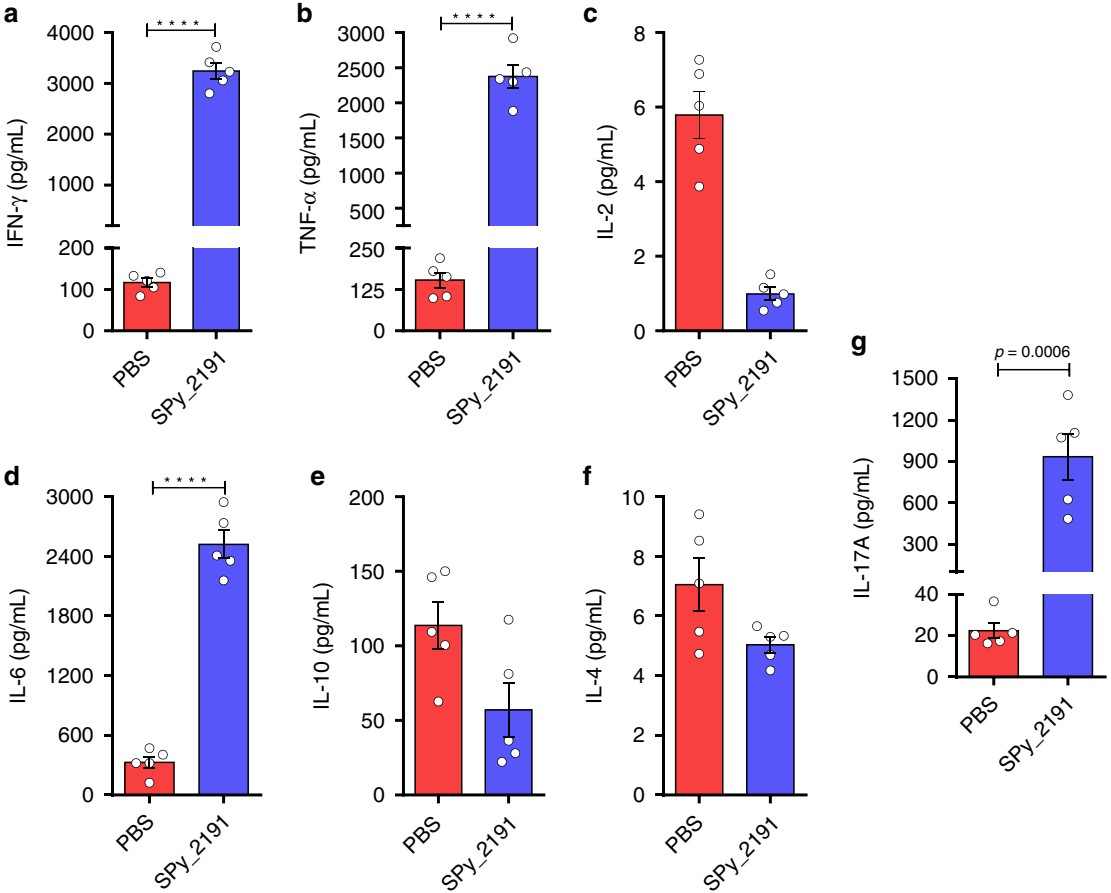

**Fig. 3 Th1-baised antigen recall response by SPy_2191-immunized mice splenocytes.** Two weeks after the final immunization, splenocytes were extracted ($n = 5$), cultured under ex vivo conditions and stimulated with SPy_2191 antigen for 72 h. **a–g** Release of Th1-specific IFN-γ, TNF-α, and IL-2; Th2-specific IL-4, IL-6, and IL-10, and Th17-specific IL-17A cytokines, was measured in the extracellular culture supernatant. Two-tailed unpaired $t$-test was employed for calculating the significant difference in all the cases. Data are presented as mean values ± SEM. ****$P < 0.0001$.

serotype coverage in parallel to the findings of opsonophagocytic killing assays; reaffirm the potential of SPy_2191 as a vaccine candidate against GAS infection.

GAS being the most common cause of bacterial pharyngitis, we further monitored the efficacy of SPy_2191 vaccination in preventing a mucosal infection. For pharyngeal colonization, GAS M9 serotype (UK) was used. For this purpose, viable counts were enumerated on day 5 post nasal infection. Interestingly, the SPy_2191-vaccinated mice showed (~19 folds) significantly lower CFUs in nasopharynx as compared to PBS-treated mice (Fig. 4f; $p < 0.01$).

**SPy_2191 provides cross-serotype protection against GAS infection.** Further, we have tested if SPy_2191 vaccination could increase the survival efficacy of mice challenged with GAS serotypes of different geographical regions. We observed that all the PBS-treated mice succumbed to death within 8 days post-challenge, regardless of the GAS serotype (Fig. 5). Most importantly, SPy_2191 was found to be significantly protective with ~92, ~86, ~86, ~81, and ~76 % mice survival after challenge with M3.1/IL, M1/IND, M1/USA, M49/IND, and M1/UK GAS serotypes, respectively ($p < 0.001$) (Fig. 5). As SPy_2191 triggered a cross-serotype immunity, it could be employed as a universal GAS vaccine candidate, protecting against prevalent and invasive serotypes from different geographical areas.

## Discussion

GAS causes high mortality and morbidity in developing countries as compared to developed countries[34]. In addition to their heterogenic serotype distribution worldwide, GAS has an array of virulence factors, which can evade or inhibit human immune system making it difficult to design a universal vaccine against all serotypes. Surface/secretory proteins were suggested as compelling vaccine candidates[19,29]. The surface-exposed M-protein of GAS has been identified as a prominent virulence factor[7] primarily due to its crucial role in adhesion to the host cell surface[35]. Therefore, M-protein-based vaccine preparation has been well established. The 30-valent vaccine was shown to evoke bactericidal antibodies against 30 vaccine serotypes and also against 24 of 40 tested non-vaccine serotypes of GAS, suggesting that its protective efficacy would be better than predicted[10]. Further, it has been reported that 26-valent M protein-based vaccine would cover more serotype diversity in developed countries but its coverage would be very less in severely affected areas and technologically less advance countries like Sub-Saharan African countries and Pacific region[36]. However, due to the emergence of new *emm* types, it has been suggested that M-protein-based multivalent vaccine should also be tested against serotypes of high-income as well as low- and moderate-income countries. Therefore, this preparation may not be effective against many unknown GAS serotypes[37]. Henceforth, there is an urgent need to identify a universal vaccine candidate against GAS infections.

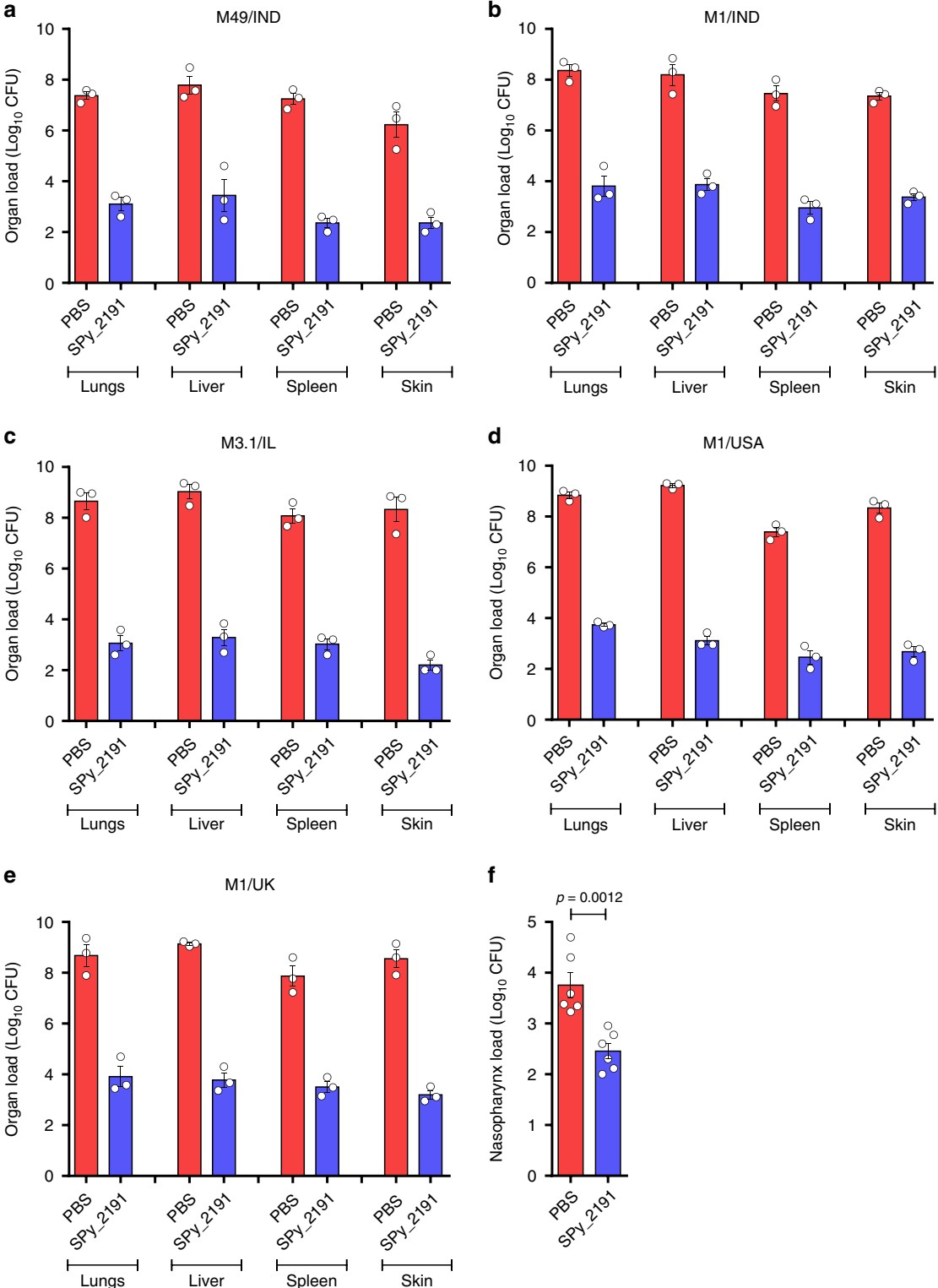

**Fig. 4 SPy_2191 immunization reduced organ load post-challenge with different GAS serotypes.** Two weeks after the final booster immunization with SPy_2191, C57BL/6 mice were challenged intraperitoneally with different GAS serotypes. Four days post-challenge, three mice from both SPy_2191-immunized and placebo groups were sacrificed, and the bacterial load (represented as $Log_{10}$ CFU) was enumerated in the liver, lungs, spleen and skin. **a**–**d** and **e** represents data of different organ load after challenge with M49/IND, M1/IND, M3.1/IL, M1/USA, and M1/UK serotypes, respectively. Red bar represents CFUs obtained from PBS control mice and blue bar represents CFUs obtained from SPy_2191-immunized mice. In case of all the organs, across all the infected serotypes, SPy_2191 effectively reduced the bacterial load compared to the placebo group (minimum significance $p < 0.01$). **f** The nasopharyngeal streptococcal load in SPy_2191 and PBS-immunized mice ($n = 6$), 5 days post-intranasal challenge with GAS M9/UK (**$P < 0.01$). Data are presented as mean values ± SEM. The level of significance was checked by using two-tailed unpaired $t$-test.

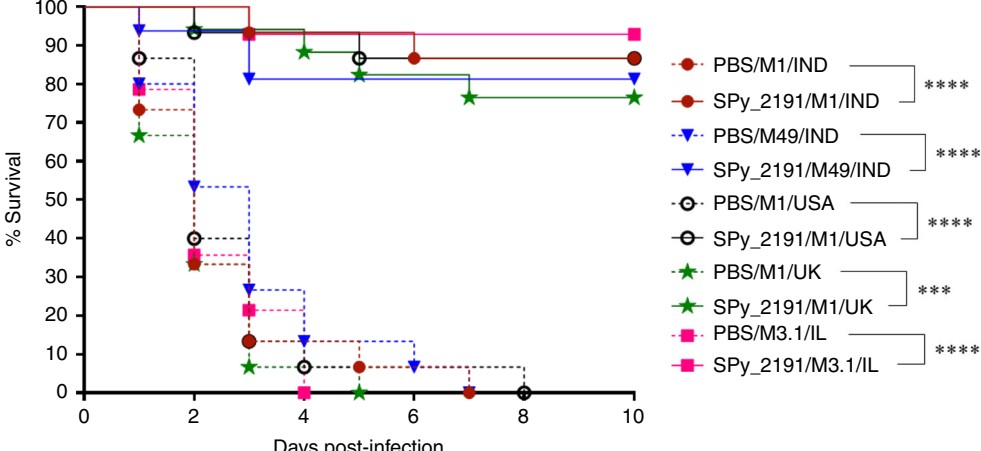

**Fig. 5 SPy_2191 immunization provided cross-serotype protection against GAS.** Survival of mice ($n = 12$) from each group infected with different GAS serotypes, was monitored for death until 10 days. The results are presented as percent survival. To calculate the significance of difference in survival studies, the log-rank test (Mantel-Cox) was applied. ***$P < 0.001$; ****$P < 0.0001$. Survival curves were generated from two independent experiments.

Identification and characterization of GAS vaccine candidates have been reported earlier also. In the past, high throughput strategies have been used to identify antigens that are immunogenic during GAS infection. Similarly, proteomic analyses of surface-associated proteins have been employed to map for suitable GAS vaccine candidates[20]. In the latter studies, interpretations have been drawn either based on antigen's immunogenicity index or on its cell surface-exposure. In order to identify a universal vaccine candidate, we have undertaken a very robust three point criteria for selection, viz. (1) candidate should be surface exposed and conserved across prevalent and disease causing GAS serotypes of developed as well as developing countries (2) involved in both GAS adhesion as well as invasion (3) should be immunogenic in nature to induce adaptive immune response in host. Based on these criteria and stringent screening, we found that SPy_2191 was the only surface-exposed required for both GAS adhesion as well as invasion (Table 1 and Fig. 1). Further, the presence and conservativeness of SPy_2191 was checked in all the available sequenced genomes of GAS. We found that it was 98% conserved in all available GAS genomes. On the basis of its conserved nature, we speculate that SPy_2191 might be essential for viability or persistence of most of the GAS serotypes circulation in human population. Additionally, we found that it induces good immunity and provides good protection in mice. Therefore present study conclusively defines a robust and improved approach to identify and validate one of the best possible universal vaccine candidates.

SPy_2191 was first reported in a genome-wide study and found to be expressed more during phagocytosis[38]. The 204 amino acid residue protein harboured a 41-mer signal peptide and was predicted to be transported by the Sec translocon and cleaved by signal peptidase I (SignalP-5.0). Further, the conserved domain prediction tool of NCBI suggests SPy_2191 to be an apparent member of the Lyz-like superfamily with a lytic transglycosylase domain and two membrane-spanning segments, similar to that of soluble lytic transglycosylases (SLT), which catalyze the cleavage of the β-1,4-glycosidic bond between N-acetylmuramic acid (MurNAc) and N-acetyl-D-glucosamine (GlcNAc). Although, at this stage the above stated functionality of SPy_2191 is only speculative and requires further biochemical characterization.

To test whether the SPy_2191 could induce an immune response in mice, after the second and third immunization, antisera were analyzed to check whether antibodies were generated against SPy_2191. We found that SPy_2191 is capable of inducing humoral adaptive immune responses as we observed significantly high IgG antibody endpoint titer after second and third immunization in comparison to first immunization (Fig. 2b). Five prevalent and invasive GAS serotypes of Israel, UK, USA, and India were tested for bactericidal assay to validate the functionality of the SPy_2191-induced humoral immunity. The average killing was observed against all tested serotypes, which range between 71–90% (Fig. 2c). Our data suggest that antibodies present in sera were able to opsonize and kill all the GAS serotypes successfully.

It is known that cytokines are the effector molecules that control and determine the type of adaptive immune response generated within a host, upon encountering any pathogen[39]. Re-stimulation of SPy_2191-vaccinated mice splenocytes with the antigen showed higher secretion of Th1-type IFN-γ and TNF-α cytokines, signifying the activation of cell-mediated adaptive immunity (Fig. 3a). TNF-α is known to instigate the NF-κB pathway which involved induction in the synthesis of other innate cytokines like IL-10 and IL-6[40]. In our study also, a significant increase in Th2-type IL-6 cytokine was observed upon SPy_2191 vaccination. Similarly, upon antigen recall, the subsequent increase in the secretion of proinflammatory cytokine-IL-17A, could be another salient arsenal of SPy_2191 immune response in increasing the protective efficacy against GAS serotypes (Fig. 5). The pivotal role of IFN-γ and IL-17A cytokines in protection and clearance of invasive GAS infections is well documented[10,41–44]. Indeed, IFN-γ at the infection site is thought to be critical for protection and it further triggers the activation and functioning of myeloid cells including polymorphonuclear leukocytes (PMNs), which play a key role in survival from GAS infections[41]. Altogether, we suggest that higher production of Th1-specific IFN-γ and TNF-α, and Th17-specific IL-17A cytokines by splenocytes in case of SPy_2191-vaccinated mice after antigen re-stimulation mimics its potentiality for a memory-recall responses against GAS infections and validates immunomodulatory competency. Whether it is mediated by CD8[+] cytotoxic T lymphocytes and CD4[+] Th1 lymphocytes is not studied in the present study[45], therefore, needs warrant investigation. Importantly, SPy_2191 was found to evoke both humoral as well as cell-mediated adaptive immune responses in mice (Figs. 2b, c, 3a, b).

Multi-organ failure during GAS infections (STSS and sepsis) can be correlated with the bacterial burden of peripheral organs. Therefore, we have also investigated whether vaccination with

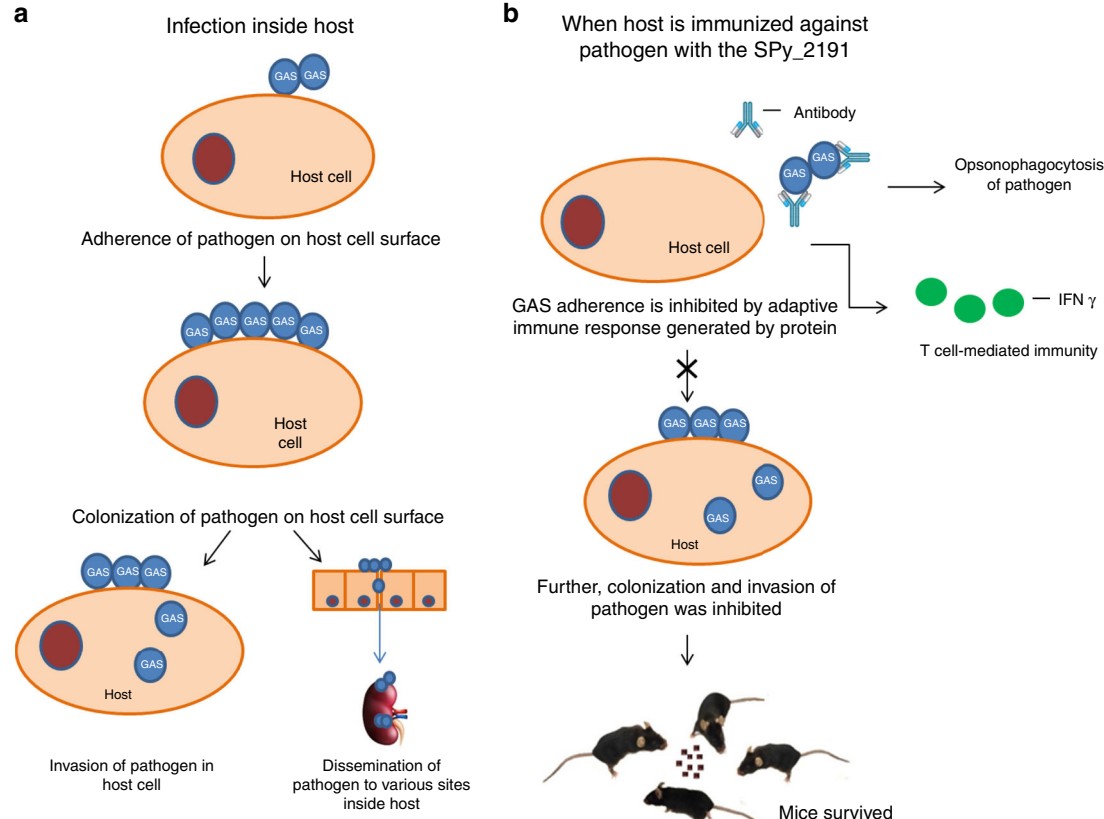

**Fig. 6 Adhesion and invasion of GAS in placebo and SPy_2191-vaccinated mice. a** Graphical representation of how GAS infects a host by first adhering then colonizing on the host cell surface, later invades and disseminates inside the cell. **b** In SPy_2191-vaccinated mice, GAS infection was bypassed plausibly due to the presence of bactericidal antibodies that caused opsonophagocytosis, thus blocking adhesion of bacteria on the host cell surface and subsequently hindered colonization and invasion of GAS within the host. Further a Th1-biased immune response and immuno-stimulation triggered by SPy_2191 helps the mice to survive against GAS challenge'.

SPy_2191 can reduce bacterial load in different organs of mice. We found that application of SPy_2191 led to enhanced bacterial clearance in lungs, liver, spleen and skin of mice upon challenge with different GAS serotypes (Fig. 4a–e). It is important to note that GAS causes ~600 million new cases of pharyngitis each year[1]. It has been reported that pharyngeal infection not only causes acute illness but also can trigger the postinfectious syndromes of poststreptococcal glomerulonephritis and acute rheumatic fever[46]. Although rheumatic fever was found to be uncommon in most developed countries, however, it is a leading cause of acquired heart disease in the school going children in case of many economically less developed countries like sub-Saharan Africa, India, etc[1]. Therefore, we have also used nasopharyngeal infection model to test SPy-2191 as a vaccine candidate. Interestingly, SPy_2191 vaccination significantly reduced GAS nasopharyngeal load upon intranasal challenge (Fig. 4f). We suggest that SPy_2191 can also be used as a vaccine candidate in case of pharyngeal infection. However, further investigations are required to determine the immunomodulatory mechanism of SPy_2191 induced protection against pharyngitis. The efficacy of a vaccine candidate is judged mainly by the degree of protection it provides to the recipient. Ability of SPy_2191 to modulate protective efficacy can be observed at different levels, starting from activation of Th1 and Th17-specific cytokines, generation of a repertoire of antibodies and reduction of bacterial burden in different organs of mice. Further, SPy_2191 was tested for in vivo protection and was found to be immunogenic and protective in mice against five prevalent and invasive GAS serotypes belongs to four different countries (Fig. 5). Ours is the first comprehensive

report on the vaccine potential of SPy_2191 in modulating host immune responses.

For the advancement of diseases in the host, bacteria must adhere, colonize on host surface and invade inside host by subverting host immune system with a range of virulence factors. Gram-positive bacteria such as Streptococci have evolved with diverse pathogenic factors works as adhesins to cause diseases. Therefore, we hypothesize that if we can inhibit adhesion of bacteria to host cells, then we may be able to bypass bacterial infection inside host (Fig. 6). Our study reports, SPy_2191 is a highly conserved, extracellular protein, involved in adhesion and invasion; protective, elicit bactericidal antibodies, hence, we conclude that SPy_2191 can act as a universal potential vaccine candidate against global important human pathogen GAS. This approach can be used for identification of new vaccine candidates against pathogenic bacteria especially those infect host severely after invasion like Listeria, Salmonella, and Shigella[47].

## Methods

**Bacterial strains and culture conditions**. GAS serotypes M1 and M49 from India (IND); M3.1 from Israel (IL); M1 from USA; and M1 and M9 (pharyngitis causing strain) from UK, were used in the present study. *Escherichia coli* DH5α and BL21 (DE3) strains were used for cloning and expression of recombinant proteins. GAS and *E. coli* cultures were grown in Todd Hewitt Broth (THB) and Luria-Bertani (LB) medium respectively at 37 °C at 220 rpm shaking. For plating and CFU enumeration of GAS serotypes, 5% sheep blood agar plates were used.

**Human cell line and culture conditions**. Human epithelial type 2 (HEp-2) cell line was procured from the National Centre for Cell Science, Pune, India and used for

neutralization assay. HEp-2 cell line was maintained in RPMI 1640 with 10% fetal bovine serum (FBS), at 37 °C with 5% $CO_2$.

**In vitro neutralization assay**. This assay was performed to find the role of selected surface proteins in adherence/invasion of GAS[24]. Mice were immunized with 45 different surface-exposed/secretory recombinant proteins followed by collection of antisera[20]. Polyclonal antibodies present in immunized antisera (gift from GSK vaccine, Siena, Italy) were incubated with GAS and then introduced on HEp-2 cell line. Preimmune sera was used as the negative control. Briefly, Indian GAS M49 was grown till $OD_{600}$ reaches ~0.5. Cells ($4 \times 10^7$) were collected and diluted in RPMI 1640 in the ratio of 1:10. Forty microliters of GAS dilution was mixed with 10 µl of mice sera (1:250 dilution in PBS) raised against each recombinant protein in 2.5 ml RPMI 1640 medium. The mixture was incubated for 1 h at 4 °C and 500 µl of it was added to confluent HEp-2 cells (80 %) at a multiplicity of infection (GAS/ mammalian cells) of 0.1:1 in a 24 well plate and kept at 37 °C in a 5% $CO_2$ incubator (Shellab, Cornelius, OH, USA) for 2 h. These infected monolayers were washed thrice with PBS to remove non-adherent GAS. To study invasion, adherent GAS cells were killed by treating the monolayers with 100 µg ml$^{-1}$ gentamycin and 5 µg ml$^{-1}$ penicillin containing complete RPMI 1640 media. The monolayer was then washed five times with PBS followed by addition of 0.2 ml of 0.25% trypsin-EDTA (Himedia, Mumbai, India) for 5–7 min at 37 °C in a 5% $CO_2$ incubator. To obtain the internalized GAS load, monolayer was disrupted using 0.8 ml of ice-cold 0.025% Triton X-100 followed by repeated pipetting. Further, this mixture was diluted accordingly and plated on 5% sheep blood agar plates. The plates were incubated overnight at 37 °C and GAS CFUs were counted.

To study adherence, all the conditions were kept the same as mentioned above, except in this case, no antibiotic treatment was given after the first 2 h of incubation. The number of attached GAS was calculated as total CFUs (attached and invaded) minus invaded CFUs. Each test was done in quadruplicate and the number of CFUs recovered per plate was determined. Percent inhibition of adherence was calculated using the formula = [1-mean CFUs in the presence of immunized serum/mean CFUs in presence of preimmunized mice serum] × 100.

**Flow cytometry analysis of surface proteins**. Surface exposed proteins were mapped on the GAS membrane using protein-specific polyclonal antibodies. In brief, the GAS M49 serotype was grown in THB to an $OD_{600}$ ~0.4, centrifuged at 7000 rpm and the resulting pellet was washed with PBS. Five hundred µl of bacterial cells were taken and suspended in 50 µl new-born calf serum. This mixture was incubated for 20 min at RT. From this, 100 µl of bacterial suspension was dispensed in fresh microcentrifuge tubes and mixed with 100 µl of preimmune/ immune sera (1:100 dilution in PBS containing 0.1% BSA) and further incubated for 30 min. Further, cells were washed with washing buffer (0.1% BSA in PBS) and incubated with goat anti-mouse IgG conjugated with FITC (in PBS containing 0.1% BSA and 20% new-born calf serum) to a final dilution of 1:100 for 30 minutes. Cells were finally washed, re-suspended in PBS and analyzed with BD FACSCalibur™ flow cytometer using BD CellQuest™ Pro software (Becton Dickinson).

**Molecular cloning, expression and purification**. SPy_2191 was PCR amplified using the genomic DNA of M49 GAS as template and gene specific primers (FP-5′CGCGGATCCATGTTTAAGAAAGAAAATTTAAAACAACG3′ and RP-5′ CCGCTCGAGGTAACCCCAAGCTGATAAACCTTG3′), at an annealing temperature of 54.7 °C. Amplified PCR product was eluted using gel extraction kit (Agilent Technologies, USA) and cloned in protein expression vector pET-21a (Novagen) at the BamHI and XhoI sites, followed by transformation into E. coli DH5α. Positive clones were screened by colony PCR using gene specific primers (described above) and confirmed by sequencing (Supplementary Fig. 2).

Plasmid (containing the cloned fragment) was isolated from the positive clones and transformed into chemically competent E. coli BL21(DE3) cells for SPy_2191 expression. The His$_6$-tagged SPy_2191 was purified by the Ni-NTA affinity chromatography. To do this, 500 ml of LB media containing 100 µg ml$^{-1}$ ampicillin was inoculated with 0.1% of overnight grown culture and grown to $OD_{600}$ ~0.6 at 37 °C with 180 rpm shaking. For protein expression, the culture was induced with 1 mM IPTG and further incubated for 4 h at 37 °C. Cells were harvested by centrifugation and suspended in extraction buffer A [50 mM Tris-HCl, pH 8.0, 300 mM NaCl, 1 mM PMSF (serine protease inhibitor), 1 mg ml$^{-1}$ lysozyme, 0.05% Triton X-100] and incubated for 30 min at room temperature. Cells were lysed by sonication (30% amplitude, at 4 °C) on a Sonics VibraCell™ digital sonicator. The lysate was centrifuged for 20 min at $11,000 \times g$ and subjected to Ni-NTA affinity chromatography (Qiagen). The His$_6$-tagged SPy_2191 bound to the column, was washed first with buffer B [50 mM Tris-Cl (pH 8.0), 300 mM NaCl and 5% glycerol] and then by buffer C (buffer B containing 30 mM imidazole). Further, protein was eluted in Buffer B containing 300 mM imidazole. The eluted fractions were analysed on SDS–PAGE and concentrated to 5 mg ml$^{-1}$ by using Macrosep® Advance Centrifugal Devices (3 kDa cut-off), dialyzed against 1xPBS pH 7.4 and stored at 4 °C.

**Mice immunization**. Six weeks old, specific-pathogen-free, inbred, female C57BL/6 mice were purchased from Hylasco Biotechnology India Pvt. Ltd., housed in a ventilated animal caging system, fed pelleted diet and water ad libitum. Mice were

immunized subcutaneously ($n = 18$) with optimal concentration i.e. 10 µg of recombinant SPy_2191 (Supplementary Fig. 5) in PBS emulsified with complete Freund's adjuvant (CFA) in the ratio of 1:1. Additionally, mice were administered two subcutaneous booster doses of SPy_2191 with incomplete Freund's adjuvant (1:1) on day 14 and 28. In the placebo group, mice were immunized with CFA/IFA in emulsification with PBS alone. Blood was collected by retro-orbital bleeding on specific time intervals (described below), sera was isolated and stored at −80 °C. Two weeks after the final booster, five mice from each group were euthanized under anaesthesia by cervical dislocation for splenocyte culturing and cytokine profiling.

**Measurement of SPy_2191-specific antibody titre**. To evaluate SPy_2191 antigen's vaccine potential, total serum IgG levels were measured on day 0 (pre-immune), 14, 28, and 42, by indirect enzyme-linked immunosorbent assay (ELISA). Briefly, 96-well high-binding, polystyrene microtiter plates (NuncMax-iSorp™) were coated with SPy_2191 at a concentration of 500 ng per well, in coating buffer containing 50 mM sodium bicarbonate (pH 9.6). The plate was incubated overnight at 4 °C, followed by blocking with 150 µl of 2% bovine serum albumin (BSA) for 2 h at 37 °C. Serial dilution of the sera was prepared in PBS ($10^2$–$10^6$) and 0.1 ml of each dilution was added to the SPy_2191 precoated wells in triplicates and further incubated for 1 h at 37 °C. The plate was then washed four times with 1×PBST (1×PBS containing 0.05% tween-20) and incubated with horseradish peroxidase (HRP) conjugated goat anti-mouse IgG at a dilution of 1:5000, in PBS containing 2% BSA. After washing four times with 1×PBST, the microtiter plate was incubated in dark with BD OptEIA TMB substrate (BD Bioscience) for 20–30 min. The reaction was stopped by adding 1 N HCl before measuring the absorbance at 450 nm using 570 nm as the reference wavelength, in a TECAN Sunrise™ microplate reader. The endpoint titre was defined as the highest sera dilution that gave an absorbance > mean+3 SD absorbance obtained at 1:100 dilution of the only PBS-immunized mice group.

**Bactericidal assay**. This in vitro assay was used to validate the functionality of SPy_2191-specific antibodies in GAS killing. For this purpose, GAS serotypes were grown at 37 °C to $OD_{600}$~0.6 in 5 ml THB broth and diluted to $10^{-5}$ in sterile 1xPBS. In each case, 50 µl of heat-inactivated serum (1:16 in dilution) was mixed with diluted GAS and incubated for 20 min at 25 °C. As a source of blood cells and complement, 400 µl of non-opsonic heparinized donor blood was added to the mixture and further incubated at 37 °C for 3 h with an end to end rotation. Each tube was serially diluted and the mixture was plated in duplicate on 5% sheep blood agar plate. Further, plates were incubated overnight at 37 °C and the resulting colonies were counted. Percent reduction in CFUs was counted by the formula— Opsonic activity of serum sample = [1-mean CFUs in the presence of SPy_2191-immunized serum/mean CFUs in presence of PBS-immunized mice serum] × 100.

**Splenocyte culturing and stimulation**. Two weeks after the final booster immunization, five mice from each group were sacrificed and their spleen was aseptically isolated. The spleen was washed with sterile PBS and crushed between frosted slides in RPMI 1640 medium containing 10% FBS. To obtain a tissue-free homogenous cell-suspension, the splenic contents were filtered through a 70 µm cell strainer. The contaminating red blood cells (RBCs) were lysed using 0.9% $NH_4Cl$, for 10 min at room temperature. The resulting cell pellet was washed three times with complete RPMI 1640 medium and cell viability was measured by the trypan blue (0.4%) staining using a haemocytometer. Subsequently, splenocytes were seeded at $10^6$ cells per well in a 96-well tissue culture plate in complete RPMI 1640 medium and stimulated with either 10 µg of SPy_2191 antigen in PBS or PBS alone (0.2 µm filter-sterilized). Cells were stimulated with concanavalin A as a positive control. Plates were incubated at 37 °C with 5% $CO_2$ and 95% humidity. After 72 h of incubation, culture supernatants were collected and stored at −80 °C until further use.

**Cytokine measurements**. The BD™ Cytometric Bead Array (CBA) mouse Th1/ Th2/Th17 cytokine kit (BD Biosciences) was used to determine the levels of Th1-specific (IFN-γ, TNF-α, IL-2), Th2-specific (IL-4, IL-6, IL-10) and Th17-specific (IL-17A) cytokines in the splenocyte culture supernatant. All the samples were processed according to the manufacturer's instructions. The data was acquired on a BD FACSCanto™ II flow cytometer (Becton Dickinson) installed at the BD-JH FACS Academy, Jamia Hamdard, India and analysed using the FCAP Array™ software, V3.0 (Becton Dickinson).

**GAS infection and protective efficacy**. All the GAS challenge experiments were carried out within laminar flow safety enclosures in the BSL-3 facility and infected animals were housed in micro-isolator cages. Mice were challenged intraperitoneally with LD50 of the desired GAS serotypes ($1 \times 10^8$ CFU per mice in case of M49/IND, M3.1/IL and M1/USA; and $5 \times 10^7$ CFU per mice in case of M1/IND and M1/UK) in PBS, 2 weeks after the final booster. For the next 10 days, mice were monitored for behavioural changes and survival or death due to GAS infection. After 10 days, all the remaining mice were euthanized.

**Measurement of bacterial load**. For enumeration of GAS load in different organs, *viz.* lungs, spleen, liver and skin, three mice from both SPy_2191-immunized and placebo groups were euthanized under anaesthesia by cervical dislocation. All the organs were collected aseptically, weighed, washed, and homogenized in 2 ml of sterile PBS. The resulting homogenate was serially diluted to $10^{-6}$ in PBS and the dilutions were plated in duplicate on 5% sheep blood agar plates. The plates were incubated overnight at 37 °C and colonies were counted next day.

**Intranasal infection**. For pharyngeal colonization, SPy_2191-immunized mice ($n = 6$) were intranasally infected with pharyngitis causing M9 (H728) strain[48]. Briefly, mice were restrained by physical means and 20 μl of droplet containing ~$1.2 \times 10^9$ CFUs was placed on the nares for inhalation. After 5 days, the naso-pharynx was aseptically dissected, weighed and homogenized in sterile PBS. Bacterial burden was monitored by plating the serial dilutions of homogenized nasal tissue onto 5% sheep blood agar plates.

**Ethical statement**. Mice infection studies were carried out in strict accordance with the recommendations given by the Institutional Animal Ethics Committee (IAEC), Jawaharlal Nehru University and Council for the Purpose of Control and Supervision of Experiments on Animals (CPCSEA), Ministry of Social Justice and Empowerment, Government of India), New Delhi. Protocols were approved by IERB board and all efforts were made to minimize the suffering of mice employed in the study with IAEC code 10/2017.

**Statistics and reproducibility**. GraphPad Prision 6 software (La Jolla, CA) was used for statistical analysis. The GAS challenge experiments were evaluated using Kaplan–Meier survival estimates and statistically analysed by using Log-rank (Mantel–Cox) test. For comparisons between multiple groups, one-way analysis of variance (ANOVA) followed by Tukey's multiple-comparison test was used. The level of significance in cytokine production and CFU enumeration studies, was checked by using two-tailed unpaired *t*-test. All attempts at replication of the data were successful. All the experiments were replicated and performed independently at least twice.

**Reporting summary**. Further information on research design is available in the Nature Research Reporting Summary linked to this article.

## Data availability

The data that support the findings of this study are available within the article and its Supplementary Information files, or are available from the corresponding author upon request. Source data are provided with this paper.

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

## Acknowledgements

P.S. is thankful to Council of Scientific and Industrial Research (CSIR), Govt. of India for providing research fellowship. M.G. is thankful to the University Grants Commission (UGC), Govt. of India, for providing Dr. D. S. Kothari Postdoctoral fellowship [No. F.4-2/2006(BSR)/17-18/0044]. A.K.J. and E.H. are thankful to the Indo-Israel project F.No:6-5/2016 (1 C) funded by UGC, Govt. of India and Israel Science Foundation. We are also very thankful to Prof. Michael Wessels, Boston children's hospital, Boston and Prof. Shiranee Sriskandan, Imperial college, London for providing GAS M1 and M9 serotypes respectively. We are very grateful to Dr. Rino Rappuoli and Dr. Immaculada, GSK vaccine, Siena, Italy for providing the recombinant sera, critical reading of the MS that helped in improving the MS.

## Author contributions

Project initiated by A.K.J., E.H., and R.B. P.S., M.G., and V.K.S. have performed the experiments. P.S., M.G., A.K.J., M.D., and V.Y. have designed the experiments. MS is written by P.S., M.G., A.K.J., R.B., and E.H. A.K.J., P.S., M.G., and A.S. analyzed the data. Chemicals were provided by A.K.J., R.B., M.D., and V.Y. Project was supervised by A.K.J. and R.B.

## Competing interests

The authors declares no competing interests.
