## [Peer Review File · Nature Communications]

Reviewers' Comments:

Reviewer #1:

Remarks to the Author:

The authors have identified an antigen of interest, based on reverse vaccinology approach, for the development of a universal vaccine against Group A Streptococcus. Efficacy is demonstrated in several clinically relevant strains through in-vitro assays (opsonophagocytic killing) and challenging immunized mice with the virulent GAS serotype M49.

The manuscript is well written, concise and experiments undertaken at a high level of proficiency for which the authors should be commended. To strengthen the conclusions of the paper, several aspects of the manuscript needs to be significantly revised, which are detailed below:

- GAS is the most common bacterial cause of acute pharyngitis following colonization of the upper respiratory tract, and RF and RHD are linked to primary pharyngeal infections (doi: 10.1136/hrt.2007.118810). What is the efficacy of Spy_2191 in preventing mucosal GAS infection in animal models?
- It is hard to ascertain the efficacy of Spy_2191 as a universal vaccine candidate in-vivo when a single challenge against M49 is presented. Undertaking challenge experiments with other clinically relevant strains as per the bactericidal assays would support the conclusion in the manuscript for universal vaccine candidate protecting against serotypes from different geographical areas.
- The reverse vaccinology approach utilized in this paper is valuable to identify cell surface antigens. However similar characterization and identification of GAS vaccine candidate proteins through analysis of the group A Streptococcus surface proteome have yielded other candidates of interest (DOI: 10.1038/nbt1179), which have not progressed any further. Highlighting why the current approach would yield more success in developing GAS vaccine candidate would provide more impact to the utility of this approach.
- The role of IFN γ levels on systemic protection can be elucidated further to highlight its importance. Several studies are suggested to the authors (DOI: 10.1086/315281, doi: 10.1038/ncomms1677).
- Animal restricted Freund's complete adjuvant (FCA) is used for inducing immune responses. A more appropriate human approved adjuvant, e.g. Alum would give more clinically relevant immune response and/or antigen-specific cytokine profile.

It was a pleasure to read and review this manuscript; authors have identified a promising GAS antigen which with further work can potentially be a promising vaccine candidate.

Reviewer #2:

Remarks to the Author:

The authors employed a reverse vaccinology approach to select an extracellular/surface-exposed protein as a promising vaccine candidate against *S. pyogenes*. This protein was the finalist due to being surface exposed and widely conserved across *S. pyogenes*. Mouse antiserum against Spy2191 inhibited adhesion and invasion in assays utilizing HEp-2 cells. Further work demonstrated that Spy2191 is immunogenic and protective in mice models, and elicits opsonophagocytic killing of diverse GAS strains in bactericidal assays. The elicitation of an IFN γ T cell response in mice was demonstrated. There has been considerable supportive work to support the author's claims, however

the manuscript organization and writing needs a great deal of attention.

Points to consider.

1. The abstract is not very informative regarding the work that was performed and discussed. For example, what exactly is being discussed regarding 9 candidates "not showing significant inhibition in invasion"?
2. Lines 88-89, results. Not a major point, but GAS strains within given serotypes can be extremely diverse. This is especially true within partern E/sof-positive strains such as those of type emm49. Within the US, the major emm49 clonal complex (which may be totally unrelated to the Indian emm49) is not particularly virulent and appears to target a specific susceptible population (homeless and or IV drug users).
3. Lines 91-96. The reader has no idea of the context of the adherence and invasion assays except through looking at the table. Need a little information within the text here.
4. lines 169-173. Would probably be better to cite the current 30 valent type-specific M vaccine as well as its ability to crossprotect against some non-vaccine M types.
5. Lines 184-185. I do not think that this statement has been proven. The term "required" for both adesion and invasion would be more apt.
6. Lines 211-216: The authors should have expanded this discussion point: Blasting the translated spy2191 sequence against the NCBI database reveals that this protein is within a conserved and widely dispersed family that contain a "lytic transglycosylast (LT) and goose egg-white lysozyme (GEWL)-like domain". In essence, this appears to be an enzyme involved in peptidoglycan maintenance. I bet that one can find a homologous counterpart in all peptidoglycan-containing bacteria. *S. equisimilis* has a protein with ~72% identity to Spy2191.
7. Lines 215-216: I would guess that Spy2191 is not only required for GAS to cause disease, but is an essential protein for cell viability.
8. Lines 216-218. I cannot understand the sentence.

Point to Point Response to Reviewers' Comments

Reviewer #1 (Remarks to the Author):

The authors have identified an antigen of interest, based on reverse vaccinology approach, for the development of a universal vaccine against Group A Streptococcus. Efficacy is demonstrated in several clinically relevant strains through in-vitro assays (opsonophagocytic killing) and challenging immunized mice with the virulent GAS serotype M49.

The manuscript is well written, concise and experiments undertaken at a high level of proficiency for which the authors should be commended. To strengthen the conclusions of the paper, several aspects of the manuscript needs to be significantly revised, which are detailed below:

Query: - GAS is the most common bacterial cause of acute pharyngitis following colonization of the upper respiratory tract, and RF and RHD are linked to primary pharyngeal infections (doi: 10.1136/hrt.2007.118810). What is the efficacy of SPy_2191 in preventing mucosal GAS infection in animal models?

Answer: We agree with the referee. Therefore, we have conducted new experiment related to this and found that SPy_2191 significantly reduces GAS pharyngeal colonization in mouse model. For this purpose we have used a pharyngitis causing GAS serotype from UK (M9). Kindly see Figure 4F in the result section. Further, we have incorporated related text in the

Results (lines 175-180)

GAS being the most common cause of bacterial pharyngitis, we further monitored the efficacy of SPy_2191 vaccination in preventing a mucosal infection. For pharyngeal colonization, GAS M9 serotype (UK) was used. For this purpose, viable counts were enumerated on day 5 post nasal infection. Interestingly, the SPy_2191-vaccinated mice showed (~19 folds) significantly lower CFUs in nasopharynx as compared to PBS treated mice (**Fig. 4F**; $p < 0.01$).

Discussion (lines 273-284). Also we have added new reference no 46.

It is important to note that GAS causes approximately 600 million new cases of pharyngitis each year¹. It has been reported that pharyngeal infection not only causes acute illness but also can trigger the postinfectious syndromes of poststreptococcal glomerulonephritis and acute rheumatic fever⁴⁶. Though rheumatic fever was found to be uncommon in most developed countries, however, it is a leading cause of acquired heart disease in the school going children in case of many economically less developed countries like sub-Saharan Africa, India, etc¹. Therefore, we have also used nasopharyngeal infection model to test SPy_2191 as a vaccine candidate. Interestingly, SPy_2191 vaccination significantly reduced GAS nasopharyngeal load upon intranasal challenge. We suggest that SPy_2191 can also be used as a vaccine candidate in case of pharyngeal infection. However, further investigations are

required to determine the immunomodulatory mechanism of SPy_2191 induced protection against pharyngitis.

Methods (lines 462-468)

Intranasal infection

For pharyngeal colonization, SPy_2191-immunized mice (n=6) were intranasally infected with pharyngitis causing M9 (H728) strain as described⁴⁸. Briefly, mice were restrained by physical means and 20 µl of droplet containing $\sim 1.2 \times 10^9$ CFUs was placed on the nares for inhalation. After 5 days, the nasopharynx was aseptically dissected, weighed and homogenized in sterile PBS. Bacterial burden was monitored by plating the serial dilutions of homogenized nasal tissue onto 5% sheep blood agar plates.

Query: - It is hard to ascertain the efficacy of Spy_2191 as a universal vaccine candidate *in-vivo* when a single challenge against M49 is presented. Undertaking challenge experiments with other clinically relevant strains as per the bactericidal assays would support the conclusion in the manuscript for universal vaccine candidate protecting against serotypes from different geographical areas.

Answer: We agree with the referee. This is an excellent and very significant suggestion. Therefore, to ascertain SPy_2191 as a universal candidate, efficacy of SPy_2191 vaccination was also tested against other prevalent and invasive GAS serotypes of USA (M1), Israel (M3.1), UK (M1) and India (M1 and M49). We found that Spy_2191 is significantly effective against all the serotypes tested. Accordingly, we have incorporated necessary text related to this in the

Abstract (lines 31-32)

SPy_2191 conferred ~76-92% protection upon challenge with these invasive GAS serotypes.

Results (lines 181-190)

SPy_2191 provides cross-serotype protection against GAS infection

Further, we have tested if SPy_2191 vaccination could increase the survival efficacy of mice challenged with GAS serotypes of different geographical regions. We observed that all the PBS treated mice succumbed to death within 8 days post-challenge, regardless of the GAS serotype (**Fig. 5**). Most importantly, SPy_2191 was found to be significantly protective with ~92, ~86, ~86, ~81 and ~76 % mice survival after challenge with M3.1/IL, M1/IND, M1/USA, M49/IND and M1/UK GAS serotypes, respectively ($p < 0.001$) (**Fig. 5**). As SPy_2191 triggered a cross-serotype immunity, it could be employed as a universal GAS vaccine candidate, protecting against prevalent and invasive serotypes from different geographical areas.

Figure 5

Discussion (lines 288-292)

Further, SPy_2191 was tested for *in vivo* protection and was found to be immunogenic and protective in mice against five prevalent and invasive GAS serotypes belongs to five different countries (**Fig. 5**). Ours is the first comprehensive report on the vaccine potential of SPy_2191 in modulating host immune responses.

Methods (lines 447-454)

GAS infection and protective efficacy

All the GAS challenge experiments were carried out within laminar flow safety enclosures in the BSL-3 facility and infected animals were housed in micro-isolator cages.

Mice were challenged intraperitoneally with LD50 of the desired GAS serotypes (1×10^8 CFU/mice in case of M49/IND, M3.1/IL and M1/USA; and 5×10^7 CFU/mice in case of M1/IND and M1/UK) in PBS, 2 weeks after the final booster. For the next 10 days, mice were monitored for behavioural changes and survival/death due to GAS infection. After 10 days, all the remaining mice were euthanized.

Supplementary information.

Determination of LD50 for different GAS serotype in C57Bl/6 mice for survival assay:

For survival assay study of a selected antigen in mice to check the protection by Spy_2191, we have determined the lethal dose of different serotype of GAS. For LD50 determination, C57Bl/6 mice (n=4) were injected intraperitoneally 100 μ l bacterial suspension (in PBS) from 10^7 , 5×10^7 and 10^8 CFU. Mice survival was recorded daily for 10 days and then they were euthanized. LD50 represents number of days elapsed when percent survival is 50%. The LD50 values of each serotype at a given CFU dose in mentioned. Values in red were used for challenge studies (**Table S2**).

Table S2. Lethal dose (LD50) calculation of different GAS serotypes CFU in C57Bl/6 mice. The LD50 values of each serotype at a given CFU dose in mentioned. Values in red were used for challenge studies.

GAS serotypes	Dose (CFU/mice)	LD50 (days)
M1/IND	10^7	8
	5×10^7	3
	10^8	2
M49/IND	10^7	-
	5×10^7	9
	10^8	3
M3.1/IL	10^7	-
	5×10^7	5
	10^8	2
M1/USA	10^7	12
	5×10^7	6

M1/UK	10⁸	3
	10 ⁷	3
	5 × 10⁷	2
	10 ⁸	1

Query: - The reverse vaccinology approach utilized in this paper is valuable to identify cell surface antigens. However similar characterization and identification of GAS vaccine candidate proteins through analysis of the group A Streptococcus surface proteome have yielded other candidates of interest (DOI: 10.1038/nbt1179), which have not progressed any further. Highlighting why the current approach would yield more success in developing GAS vaccine candidate would provide more impact to the utility of this approach.

Answer: We thank the reviewer for this comment and do agree that similar identification and characterization of GAS vaccine candidates have been reported earlier also (ref 20). We have added the following text in the

Discussion (lines 210-228)

Identification and characterization of GAS vaccine candidates have been reported earlier also. In the past, high throughput strategies have been used to identify antigens that are immunogenic during GAS infection. Similarly, proteomic analyses of surface-associated proteins have been employed to map for suitable GAS vaccine candidates²⁰. In the latter studies, interpretations have been drawn either based on antigen's immunogenicity index or on its cell surface-exposure. In order to identify a universal vaccine candidate, we have undertaken a very robust three point criteria for selection, viz. (1) candidate should be surface exposed and conserved across prevalent and disease causing GAS serotypes of developed as well as developing countries (2) involved in both GAS adhesion as well as invasion (3) should be immunogenic in nature to induce adaptive immune response in host. Based on these criteria and stringent screening, we found that SPy_2191 was the only surface-exposed and involved in both GAS adhesion as well as invasion (**Table 1 & Fig. 1**). Further, the presence and conservedness of SPy_2191 was checked in all the available sequenced genomes of GAS. We found that it was 98% conserved in all available GAS genomes. On the basis of its conserved nature, we speculate that SPy_2191 might be essential for viability or persistence of most of

the GAS serotypes circulation in human population. Additionally, we found that it induces good immunity and provides good protection in mice. Therefore present study conclusively defines a robust and improved approach to identify and validate one of the best possible universal vaccine candidates.

Query: - The role of IFN γ levels on systemic protection can be elucidated further to highlight its importance. Several studies are suggested to the authors (DOI: 10.1086/315281, doi: 10.1038/ncomms1677).

Answer: We are very thankful to the reviewer for the valuable comment. As suggested, we have discussed the significant role of heightened IFN- γ levels in protection to GAS infections. For details see

Discussion (lines 257-265).

The pivotal role of IFN- γ and IL-17A cytokines in protection and clearance of invasive GAS infections is well documented^{10, 41, 42, 43, 44}. Indeed, IFN- γ at the infection site is thought to be critical for protection and it further triggers the activation and functioning of myeloid cells including polymorphonuclear leukocytes (PMNs), which play a key role in survival from GAS infections⁴¹. Altogether, we suggest that higher production of Th1-specific IFN- γ and TNF- α , and Th17-specific IL-17A cytokines by splenocytes in case of SPy_2191-vaccinated mice after antigen re-stimulation mimics its potentiality for a memory recall responses against GAS infections and validates immuno-modulatory competency.

Further four new references have been added Ref No: 41-44.

Query: - Animal restricted Freund's complete adjuvant (FCA) is used for inducing immune responses. A more appropriate human approved adjuvant, e.g. Alum would give more clinically relevant immune response and/or antigen-specific cytokine profile.

Answer: We agree with the referee's suggestion. However, the primary aim of the present study is to evaluate and establish the vaccine potential of SPy_2191 against GAS infections. As suggested by the reviewer, using FDA-approved, licenced adjuvant (suitable for human-use) like alum, is the part of our next story. After completing the present study our next target is to do such experiments in details.

Query: It was a pleasure to read and review this manuscript; authors have identified a promising GAS antigen which with further work can potentially be a promising vaccine candidate.

Answer: We are very thankful to the referee for the helpful suggestions. We found the reviewer's comments very helpful and important in improvising the MS.

Reviewer #2 (Remarks to the Author):

General Query: The authors employed a reverse vaccinology approach to select an extracellular/surface-exposed protein as a promising vaccine candidate against *S. pyogenes*. This protein was the finalist due to being surface exposed and widely conserved across *S. pyogenes*. Mouse antiserum against Spy2191 inhibited adhesion and invasion in assays utilizing HEp-2 cells. Further work demonstrated that Spy2191 is immunogenic and protective in mice models, and elicits opsonophagocytic killing of diverse GAS strains in bactericidal assays. The elicitation of an IFN γ T cell response in mice was demonstrated. There has been considerable supportive work to support the author's claims, however the manuscript organization and writing needs a great deal of attention.

Answer: Thanks for praising the work. We have revised the MS as per the referee suggestions.

Points to consider.

Query: 1. The abstract is not very informative regarding the work that was performed and discussed. For example, what exactly is being discussed regarding 9 candidates “not showing significant inhibition in invasion”?

Answer: We apologize for the same and hence we have thoroughly revised the abstract and made necessary changes. Please see the revised abstract.

Abstract (lines 21-34)

Using reverse vaccinology approach, 18 surface proteins were selected and tested as potential vaccine candidates against human pathogen group A Streptococcus (GAS). Eight candidates were neither found conserved nor surface-exposed and 9 candidates failed to significantly inhibit GAS invasion, hence these 17 candidates were rejected. Only SPy_2191 was found to be conserved, surface-exposed and inhibited both GAS adhesion and invasion. SPy_2191 mice immunization generated bactericidal antibodies that endorsed ~70-85% opsonophagocytic killing of prevalent and invasive GAS serotypes of different geographical region, viz. M1 and M49 (India), M3.1 (Israel), M1 (UK) and M1 (USA). Resident splenocytes showed higher interferon- γ and tumour necrosis factor- α secretion upon antigen re-stimulation, suggesting activation of cell-mediated immunity. SPy_2191 significantly reduced streptococcal load in the organs. Importantly, SPy_2191 conferred ~76-92% protection upon challenge with these invasive GAS serotypes. Further, it significantly suppressed GAS pharyngeal colonization in mice mucosal infection model. Our findings suggest that SPy_2191 can act as a universal vaccine candidate against GAS infections.

Query 2. Lines 88-89, results. Not a major point, but GAS strains within given serotypes can be extremely diverse. This is especially true within partern E/sof-positive strains such as those of type emm49. Within the US, the major emm49 clonal complex (which may be

totally unrelated to the Indian emm49) is not particularly virulent and appears to target a specific susceptible population (homeless and or IV drug users).

Answer: We do agree to the reviewer's comment that GAS strains within a given serotypes can be extremely diverse. It is therefore to be noted that our study includes five different serotypes (M1 IND, M1 USA, M1 UK, M49 IND and M3.1 ISRAEL) from different geographical regions. Our whole purpose was to validate the potential of vaccine candidates against multiple GAS serotypes which has been successfully demonstrated. Furthermore, M1 serotype from USA has been included in this study which constitutes good proportion of all GAS serotypes.

Query 3. Lines 91-96. The reader has no idea of the context of the adherence and invasion assays except through looking at the table. Need a little information within the text here.

Answer: We apologize for the same. For reader's clarity we have added more information and made necessary changes in the

Results (lines 87-98)

Inhibition of adherence and invasion

For effective vaccination, the potential vaccine candidate must be surface exposed, involved in invasion and adherence^{23,24,25,26}. Out of 52 previously predicted vaccine candidates, 45 sets of immune and pre-immune mouse antisera, generated against recombinant surface/secretory proteins of GAS (**Supplementary Table S1**)^{20,21} were used to investigate whether the corresponding surface/secretory proteins had any role in adherence or invasion. Initially, GAS serotype M49 that caused outbreaks in India and USA was used for this study as this serotype was found to be most invasive^{27, 28}. We found that 18 out of 45 antisera inhibited GAS adhesion by $\geq 75\%$, however only 7 antisera abolished invasion by $\geq 80\%$ (**Table 1**). We found SPy_2191 antisera inhibited adherence and invasion by 89 and 93%, respectively (**Table 1**). Pre immunized mice antisera for each recombinant GAS protein were used as a control.

For more clarity a new table S1 is added in the supplementary section.

Table S1: List of 45 recombinant antisera generated against selected surface proteins used in this study.

S. No	Sample No.	Sample Name
1.	NP_268607	N-acetylneuraminate-binding proteins
2.	NP_268656	ABC transporter substrate-binding protein
3.	NP_268747	ABC transporter metal binding protein (lipoprotein)
4.	NP_268946	Zinc-binding lipoprotein AdcA precursor
5.	NP_269262	Spermidine/putescine ABC transporter periplasmic transport protein
6.	NP_269379	Phosphate ABC transporter periplasmic phosphate Plasmodium falciparum –binding protein
7.	NP_269403	Putative amino acid ABC transporter, periplasmic amino acid-binding protein
8.	NP_269961	Surface lipoprotein
9.	NP_269968	Laminin adhesion
10.	NP_269421	Maltose/maltodextrin-binding protein
11.	NP_268436	secreted protein/ Hypothetical protein SPy_0019
12.	NP_269203	Extracellular hyaluronate lyase
13.	NP_269467	Internalin A
14.	NP_269488	Foldase PrsA
15.	NP_269743	Esterase
16.	NP_269810	Hypothetical protein SPy_1801/ Immunogenic secreted precursor-like protein
17.	NP_269818	Hypothetical protein SPy_1813
18.	NP_269916	Hypothetical protein SPy_1939
19.	NP_269944	Streptokinase A/ Streptokinase A precursor
20.	NP_269976	Hypothetical protein SPy_2025/ Immunogenic secreted protein precursor
21.	NP_269981	ATP-binding cassette transporter-like protein
22.	NP_269984	Foldase PrsA
23.	NP_270099	Hypothetical protein SPy_2191/ Transglycosylase SLT domain family protein
24.	NP_269045	Hypothetical protein SPy_0836
25.	NP_268639	Penicillin binding protein (D-alanyl-D-alanine carboxypeptidase)
26.	NP_269051	Hypothetical protein SPy_0843
27.	NP_270005	Dipeptidase
28.	NP_270119	Serine protease
29.	NP_268581	Hypothetical protein SPy_0210

30.	NP_269063	Peptidoglycan hydrolase
31.	NP_269570	Hypothetical protein SPy_1492
32.	NP_269625	Hypothetical protein SPy_1558
33.	NP_269940	Pullulanase
34.	NP_268582	Exotoxin G
35.	NP_269402	cAMP factor
36.	NP_268943	Pyrogenic exotoxin C
37.	NP_268860	Hypothetical protein SPy_0604
38.	NP_269208	Hypothetical protein SPy_1037
39.	NP_269417	Hypothetical protein SPy_1290
40.	NP_269569	Hypothetical protein SPy_1491
41.	NP_268542	ABC transporter lipoprotein
42.	NP_269268	Acid phosphatase/ phosphotransferase
43.	NP_268750	Cyclophilin-type protein
44.	NP_268760	Hypothetical protein SPy_0469/ putative 42 KDa protein
45.	NP_269989	Mitogenic factor

Note: All 45 recombinant sera were kind gift of GSK vaccine, Siena, Italy.

Query 4. lines 169-173. Would probably be better to cite the current 30 valent type-specific M vaccine as well as its ability to cross-protect against some non-vaccine M types.

Answer: As suggested by the reviewer we have cited the same in the discussion section of the revised manuscript. Thank you for pointing this oversight, though we have cited the same reference in the introduction. We have added related text in the discussion. Please see

Discussion (lines 199-201)

The 30-valent vaccine was shown to evoke bactericidal antibodies against 30 vaccine serotypes and also against 24 of 40 tested non-vaccine serotypes of GAS, suggesting that its protective efficacy would be better than predicted¹⁰.

Query 5. Lines 184-185. I do not think that this statement has been proven. The term “required” for both adesion and invasion would be more apt.

Answer: Thanks for pointing this out. Accordingly we have modified the lines as follows.

Discussion (lines 219-221)

Based on these criteria and stringent screening, we found that SPy_2191 was the only surface-exposed required for both GAS adhesion as well as invasion (**Table 1 & Fig. 1**).

Query 6. Lines 211-216: The authors should have expanded this discussion point: Blasting the translated spy2191 sequence against the NCBI database reveals that this protein is within a conserved and widely dispersed family that contain a “lytic transglycosylase (LT) and goose egg-white lysozyme (GEWL)-like domain”. In essence, this appears to be an enzyme involved in peptidoglycan maintenance. I bet that one can find a homologous counterpart in all peptidoglycan-containing bacteria. *S. equisimilis* has a protein with ~72% identity to Spy2191.

Answer: We thank the reviewer for this excellent suggestion. Indeed, homology studies reveal SPy_2191 to be a member of Lyz-like superfamily. As suggested by the reviewer, the same has been elaborated in the discussion section. Please see

Discussion (lines 229-237)

SPy_2191 was first reported in a genome-wide study and found to be expressed more during phagocytosis³⁸. The 204 amino acid residue protein harboured a 41-mer signal peptide and was predicted to be transported by the Sec translocon and cleaved by signal peptidase I (SignalP-5.0). Further, the conserved domain prediction tool of NCBI suggests SPy_2191 to be an apparent member of the Lyz-like superfamily with a lytic transglycosylase domain and two membrane-spanning segments, similar to that of soluble lytic transglycosylases (SLT), which catalyze the cleavage of the β -1,4-glycosidic bond between N-acetylmuramic acid (MurNAc) and N-acetyl-D-glucosamine (GlcNAc). Although, at this stage the above stated functionality of SPy_2191 is only speculative and requires further biochemical characterization.

A new reference No: 38 also been added for this purpose.

Query: 7. The Lines 215-216: I would guess that Spy2191 is not only required for GAS to cause disease, but is an essential protein for cell viability.

Answer: Many thanks for the suggestion. We apologize for the same and hence we have reconstructed the sentence and made necessary changes. Please see

Discussion (lines 221-225)

Further, the presence and conservedness of SPy_2191 was checked in all the available sequenced genomes of GAS. We found that it was 98% conserved in all available GAS genomes. On the basis of its conserved nature, we speculate that SPy_2191 might be essential for viability or persistence of most of the GAS serotypes circulation in human population.

Query 8. Lines 216-218. I cannot understand the sentence.

Answer: In order to avoid any confusion and for the clarity we have deleted the lines as well as the reference in the revised MS.

Reviewers' Comments:

Reviewer #1:

Remarks to the Author:

The authors are to be commended on a well-written and well executed manuscript post-review. The experiments are executed well with appropriate statistical analysis. In this reviewer's humble opinion, one cannot help but think that the efficacy of the promising antigen would be more impactful if a human approved adjuvant was used instead of CFA, or a portion of the secondary challenge experiments was carried out with a more clinically relevant adjuvant. This would highlight to the reader the true promise of this antigen, especially as the authors have stated they will be carrying out further studies. I look forward to seeing those future studies.

REVIEWERS' COMMENTS:

Reviewer #1 (Remarks to the Author):

The authors are to be commended on a well-written and well executed manuscript post-review. The experiments are executed well with appropriate statistical analysis. In this reviewer's humble opinion, one cannot help but think that the efficacy of the promising antigen would be more impactful if a human approved adjuvant was used instead of CFA, or a portion of the secondary challenge experiments was carried out with a more clinically relevant adjuvant. This would highlight to the reader the true promise of this antigen, especially as the authors have stated they will be carrying out further studies. I look forward to seeing those future studies.

Answer: We thank the reviewer for praising the work. We agree with the referee's suggestion. The primary aim of the present study was to evaluate and establish the vaccine potential of SPy_2191 in protection to GAS infections. As suggested by the reviewer, using FDA-approved, licensed adjuvant (suitable for human-use) like alum, is the part of our next story. After completing the present study our next target is to do such experiments in details.